# Cutaneous Adverse Reactions Associated with Monoclonal Antibodies Treatment in Multiple Sclerosis: Case Reports and Short Literature Review

**DOI:** 10.3390/jcm11133702

**Published:** 2022-06-27

**Authors:** Carmen Adella Sirbu, Raluca Ivan, Titus Mihai Vasile, Lucian George Eftimie, Daniel Octavian Costache

**Affiliations:** 1Department of Neurology, ‘Dr. Carol Davila’ Central Military Emergency University Hospital, 010242 Bucharest, Romania; sircar13@yahoo.com (C.A.S.); ivan.raluca23@gmail.com (R.I.); 2Clinical Neurosciences Department, University of Medicine and Pharmacy “Carol Davila” Bucharest, 050474 Bucharest, Romania; 3Department of Pathology, ‘Dr. Carol Davila’ Central Military Emergency University Hospital, 010242 Bucharest, Romania; 4Department of Dermatology, ‘Dr. Carol Davila’ Central Military Emergency University Hospital, 010242 Bucharest, Romania; daniel_costache@yahoo.com; 52nd Dermatology Discipline, Faculty of Medicine, University of Medicine and Pharmacy “Carol Davila” Bucharest, 050474 Bucharest, Romania

**Keywords:** multiple sclerosis, monoclonal antibodies, natalizumab, ocrelizumab, adverse skin reactions

## Abstract

Background and aims. Multiple sclerosis is a disease of the central nervous system, whose treatment often involves the use of monoclonal antibodies. This can lead to a series of complications that the clinician should pay attention to and accordingly adjust the therapy. We aim to emphasize real-life experiences with adverse cutaneous reactions to monoclonal antibodies by presenting a series of two cases from our clinic. Methods. In the first case, a female patient was treated with natalizumab for eight years and developed relapsing-remitting cutaneous lesions following the monthly administration of the treatment. The second case is of a male patient treated with ocrelizumab, who developed plaque-like lesions following the fifth administration. We analyzed the biological parameters and performed investigations, dermatological evaluation and skin biopsies. Results. The result of the skin biopsy for the natalizumab patient showed a chronic spongiotic dermatitis, with the anti-natalizumab antibodies being negative. The patient who received ocrelizumab developed nummular eczema, disseminated on his trunk and limbs. Conclusions. Given the fact that these therapies are frequently used in multiple sclerosis patients, and their skin adverse reactions are known, we described some particularities and a brief review of the literature with practical implications. Further studies need to be conducted to establish a firm association between monoclonal antibodies therapy and adverse cutaneous reactions, but the clinician should be aware of their existence.

## 1. Introduction

A clear etiology and a particular pathological mechanism for multiple sclerosis (MS) have not yet been established and therefore its management requires further studies. In recent years, progress has been made regarding this subject, with new innovative medications depending on the type of the disease. Clinically, the disease can have a relapsing or a progressive pattern [1]. Given that the underlying pathological process is different depending on the type, certain therapies have been developed, the efficacy and level of recommendation being decided based on the MS form and other eligibility criteria. We aim to emphasize some particularities of the possible adverse cutaneous reactions to these monoclonal antibodies, more precisely for natalizumab and ocrelizumab, by presenting a series of two cases from our clinic, alongside rigorous literature research regarding the existing reported cases.

Natalizumab is a monoclonal antibody used for the treatment of RMS that targets α4β1-integrin, an adhesion molecule found on the surface of lymphocytes. This prevents their binding to the VCAM-1 endothelial receptor, thus blocking the lymphocytes inside the blood vessels and impeding the inflammation of the CNS [2]. It is administered intravenously, every 4 weeks, being highly effective in reducing relapses and slowing the progression of the disease compared to interferon beta or placebo [3,4]. Regarding adverse reactions, the most severe, but rare (approximately in 0.1% of cases), is progressive multifocal leukoencephalopathy (PML). The screening for the presence of the JC virus is necessary at the beginning of the treatment. Systemic allergic reactions have also been described, most often after a few hours from the infusion, after the administration of the second dose. This reaction has been described concerning the appearance of the neutralizing antibodies against natalizumab [5]. Other cutaneous reactions that can occur are melanomas, urticaria, allergic dermatitis, and psoriasis [6,7]. It is important to stress these adverse reactions, as they are described as frequent in the medication’s leaflet [8].

Ocrelizumab is a recombinant human monoclonal antibody, highly effective in the activity and progress of the disease [9]. It is directed against CD20 from the surface of the B cells. Therefore, it limits the invasion of B cells in the CNS from the periphery, reduces the interaction between T cells and B cells and the production of inflammatory cytokines, and it also reduces the formation of mature and active plasmacytes. The first administration is of 300 mg intravenously, followed by a second infusion of 300 mg two weeks later. After this first dose, it is given every six months, as a 600 mg intravenous infusion [1,10]. Given the fact that this is still a very new treatment, the adverse reactions have not yet been fully understood, especially in the long term. The depletion of B-cells can lead to a higher risk of infections, such as shingles, herpes simplex or varicella-zoster infections, upper or lower respiratory tract infections, or even neoplasms. There is also a risk for hypogammaglobulinemia [10,11]. It is important to document the possible adverse reactions and the impact on the patient’s quality of life, as these may influence the patient’s compliance to treatment. In the medication leaflet, cutaneous adverse reactions are reported frequently after the infusion, as infusion-related reactions. These have been encountered with a higher rate during the first infusion, or within 24 h after, in the form of pruritus, rash, urticaria, or erythema. Respiratory symptoms have also been described, as well as hypotension and hypersensitivity reactions [12]. Other cutaneous reactions, such as psoriasiform dermatitis, palmoplantar pustulosis, or oral lichen planus, have been described in the literature, but the causative relation between them and the treatment has not been fully determined [11,13].

## 2. Materials and Methods

### 2.1. Case 1

A 47-year-old female patient was diagnosed with relapsing multiple sclerosis (RMS) in 2003. In 2011, she started therapy with natalizumab, without any acute adverse reactions. She has a positive maternal family history for scleroderma, both male children are positive for psoriasis and her aunt has been diagnosed with rheumatoid polyarthritis. Eight years after the initiation of treatment, her JCV (John Cunningham Virus) status remained negative. In 2019 she reported recurrent urticarial erythema, first limited to her upper limbs, and then extending to her neck, face, and shoulders, accompanied by pruritus, sometimes with spontaneous resolution over a few days. The moment of appearance was two to four days after the monthly natalizumab infusion. At that point, she consulted a dermatologist, who recommended treatment with a topical corticosteroid (fluticasone), and two oral antihistamines (desloratadine and levocetirizine), with good outcome. The eruption continued to appear monthly, a few days to a week after every infusion, with complete resolution in several days, under the recommended treatment. The patient’s recurrent cutaneous episodes did resolve completely after a few days, some with spontaneous resolution, but others after the administration of the suggested dermatological treatment. Before the biopsy, the fleeting character of the lesions raised the probability of a histaminergic nature, supporting the use of antihistaminic medications. Their use ameliorated the patient’s skin reaction, but it should be remembered that alongside this therapy she also followed a topical corticosteroid treatment, with anti-inflammatory effects. It is also useful to mention that the periodic natalizumab infusions do not include the use of any other drug, regardless of topical corticosteroids and antihistaminic medication. The infusion room is not entirely latex-free, given the material of the gloves used by the nurses during the infusion.

Her clinical neurological examination showed a patient with positive Lhermitte sign, altered vision on the right eye, left ear hypoacusis, right central facial paresis, reduced sensation to temperature, pain, and fine touch on the left body, a sensibility level on T10, left hemiparesis with the increased muscular tone of the lower limbs (2 points on the Ashworth scale), left hand and foot ataxia, Babinski sign on the left, generally brisked tendon reflexes (on the left body more than on the right), urinary incontinence, impossible tandem walk and ambulatory for 100 m. The Expanded Disability Status Scale (EDSS) was 5.5 points.

### 2.2. Case 2

A 42-year-old male patient was diagnosed with primary progressive multiple sclerosis (PPMS) in 2019, at the age of 39 years, with no personal or familial history of autoimmune diseases. At the time of the diagnosis, he presented left hemiparesis, paresthesia, altered sensibility and ataxia on the left body, and a left Babinski sign. The treatment was initiated with ocrelizumab, without adverse reactions reported at the time of the periodic infusions and without acute hypersensitivity reactions. A couple of days after the fifth administration, in September 2021, he reported a cutaneous lesion on the right calf, well delimited, erythematous, with pruritus, and covered with scales. Over the next few weeks, similar smaller lesions appeared on his back and his arms. At that point, he presented to a dermatologist who raised the suspicion of disseminated nummular eczema and recommended treatment with topical corticosteroids, antihistamines, oral antibiotics, and topical crystal violet solution. He affirmed that after the treatment the pruritus diminished, and the red aspect faded, but they did not disappear completely. Until the present evaluation, he experienced periods of exacerbation, with pruritus, that yielded with the administration of the crystal violet solution. Given the rapid dissemination of the lesions and the immune therapy that could predispose the patient to secondary cutaneous infections, the decision to directly initiate a short course of oral antibiotic treatment was made, in absence of the mandatory signs of secondary infection. This attitude is also supported by the lesion’s recurrence and persistency [14]. He affirmed that after the treatment the pruritus diminished, and the red aspect faded, but they did not disappear completely. 

The present neurological evaluation showed intact cranial nerves, ataxia of the left limbs, normal sensibility, spasticity of the inferior limbs (grade 3 on the Ashworth scale), brisk osteotendinous reflexes (more on the left side), bilateral Babinski sign, spastic walk, possible on small distances (maximum 100 m), difficult tandem walk, positive Romberg test, and urinary incontinence and urgency. The EDSS score was 5.5 points.

## 3. Results

In the first patient, a skin biopsy was done, and the histopathological result was chronic spongiotic dermatitis (Figure 1).

Concerning the treatment recommended, according to certain studies, the topical corticosteroids have little effect on the acute urticaria, with a slight benefit on the early erythematous eruption in response to local histamine elevated levels. Despite their anti-inflammatory effects based on the vasoconstrictor properties, studies do not support the use of topical corticosteroids in confirmed acute urticaria, as they do not affect the mast cell degranulation. However, in our case, waiting for the result of the skin biopsy, the use of corticosteroids was tented also to cover other skin lesions that could benefit from this treatment (i.e., psoriasis, atopic dermatitis, eczema, discoid lupus erythematosus). After the result, the treatment was adapted accordingly, permitting the patient to control the dermatological condition and continue the multiple sclerosis treatment.

At the current hospitalization of the ocrelizumab treated patient, the cutaneous eruption was present on his right calf, his left hip, and his lower back, with a low degree of itching. We sent him to be seen again by a dermatologist, this time also with the completion of a biopsy of the lesion. The result of the skin biopsy, however, described nummular eczema (Figure 2), which is a chronic inflammatory cutaneous disease [14]. The patient is programmed to receive his periodic ocrelizumab infusion at the end of the current month, with close monitoring of the evolution of his skin lesions.

Both patients underwent allergology testing, including against latex, immunological, infectious and antineoplastic panels. The structured allergy history was negative, as well as the skin prick tests. To exclude the possibility of a delayed-type reaction, they were also advised to exclude or reintroduce certain foods from or into their diet and they have been offered dietary advice. No apparent cause was identified for the skin eruption.

## 4. Discussion

These cases aim to demonstrate that in absence of other causes for new cutaneous lesions, it is safe to think about the connection between them and the disease-modifying therapy administered.

There are several possible theories regarding the cutaneous lesions that arise after natalizumab infusions. First of all, in the drug’s leaflet, the cutaneous adverse reactions such as pruritus, rash, and urticaria are marked as common (meaning ≥1/100 to <1/10). Most of the hypersensitivity reactions, being considered type I allergic reactions, appeared in the first 60 min after the completion of the infusion [8]. Our patient however did not experience any adverse reactions in the first 60 min after the administration of treatment.

Other studies from the literature raise the possibility of a causal relation between natalizumab and the appearance of psoriatic lesions. First, this is a chronic inflammatory immune-mediated skin disease, with a yet unknown etiology. Given the common risk factors with MS (certain similar pathophysiology pathways, e.g., the involvement of the Th17 cells and genetic risk variants, IL23R polymorphisms) the association of these two diseases, with or without the addition of treatment, is very likely. In essence, as mentioned above, natalizumab, by blocking the interaction between α4β1 integrin and VCAM-1, inhibits the leukocyte transition through the BBB. In this way it raises the number of lymphocytes and CD4 +, IL-17 + cells in the blood, with the secretion of IL-17 at this level. Furthermore, it can lead to increased production of proinflammatory cytokines. This is one way of explaining a possible link between them [6]. This connection is, however, mostly described with the administration of IFN-β treatment [15].

In psoriasis, cutaneous biopsy shows the infiltration of the epidermis by Th1, Th17, γδ T cells, lymphocytes, and dendritic cells, leading to subsequent keratinocyte proliferation. In MS there is an infiltration with T cells targeted against the myelin sheaths, with a neuroinflammatory response, blood–brain barrier (BBB) destruction, and the activation of further immune cells. To this point, the data available regarding their link are scarce [6]. There is also the hypothesis that the medication can exacerbate a previous autoimmune disturbance predisposition [16]. This is a plausible track given our patient’s familial history.

Another theory regarding the possible cutaneous reaction that can occur during the treatment with natalizumab is based on the action that the drug has on integrins, heterodimers considered to be essential in the composition of the extracellular matrix. This matrix alteration, with subsequent activation of an inflammatory cascade, can lead to several dermatological conditions, such as the appearance of a cutaneous sarcoidosis-like reaction [17]. However, the dermatological examination did not sustain this possibility.

A relatively more common delayed allergic reaction described in the literature is based on the formation of anti-natalizumab antibodies. Data show that 68% of the patients treated with natalizumab who had hypersensitivity reactions were positive for these antibodies. Clinically resembling the delayed infusion reactions, and concerning the anti-natalizumab antibodies, the studies reported type III hypersensitivity reactions or serum sickness-type reactions. The formation of these antibodies is most frequent after approximately 4 weeks after a single natalizumab injection. Therefore, the clinician must take into consideration the testing of the patients who present hypersensitivity reactions that can, in some cases, lead to the interruption of the treatment. Studies have shown that type III allergic reactions are not so frequent with humanized monoclonal antibodies, compared with chimeric monoclonal antibodies [5]. Likewise, natalizumab is demonstrated to increase the number of lymphocytes, monocytes, basophils, and eosinophils in the peripheral blood; this effect is related to possible allergic reactions [18]. Our patient, however, presented normal values of eosinophils during the periodic checkups.

Studies have also linked the use of Natalizumab with some severe viral or fungal infections such as herpes simplex, varicella-zoster, mycobacteria, JC virus or Candida. These adverse reactions are common among the patients treated with monoclonal antibodies and a thorough investigation of the patients’ immune status is highly recommended [19].

Given our patient’s background and her family history, it was safe to assume that her cutaneous reaction was either a newly triggered autoimmune disease or just a common hypersensitivity reaction to her monthly medication. These assumptions were made especially when taking into consideration the macroscopic aspect, the temporal relation of the treatment with the skin eruption, and the lack of any other concomitant drug.

According to the skin biopsy, her lesions were compatible with chronic spongiotic dermatitis. This is described as a superficial inflammatory dermatosis that involves the first compartment of the skin, with changes in the epidermal and superficial perivascular inflammatory infiltrate [20]. It can be caused by various clinical conditions, with one possible pathophysiological explanation being the dysregulation of the natural killer cells (NK cells) [21].

Although this skin lesion has not yet been described, natalizumab treatment studies show that this monoclonal antibody increases the biological activity of NK cells, their cytolytic potential and the release of perforins and granzymes [22]. This can provide a plausible link between the appearance of the new eruption and the treatment.

One particularity of the cases presented in this article is the appearance of lesions following the natalizumab infusions after a few days, not in the immediate hours such as in a classic hypersensitivity reaction (within 2 h after the infusion). Usually, the delayed allergic reaction occurs in the context of the development of anti-natalizumab antibodies; the most common hypersensitivity reactions described in the existing studies were linked with the formation of these antibodies. Therefore, the measurement of these antibodies was negative [12]. However, given the fact that the eruption was not severe, and because the MS was very well managed with this treatment, the decision to continue the therapy with natalizumab was made.

As mentioned above, ocrelizumab is a relatively new accepted drug for the treatment of PPMS, with increasing use in the later years and rare reported adverse reactions. It targets the CD-20 expressing cells, thereby depleting the B-lymphocytes. Studies have shown that it slows the disease progression, both clinical and imagistic, compared to placebo [9].

The patients must be very well evaluated before the beginning of this treatment, with a final proven positive benefit/risk ratio. The most common adverse reactions reported with the use of ocrelizumab are infusion-related reactions, hypersensitivity reactions, opportunistic infections, herpetic or respiratory infections, and the reactivation of hepatitis B. There is also a small risk of PML described for the treatment with anti-CD20 antibodies, but this is also encountered in other diseases than MS [9].

Cases of psoriasiform dermatitis have been described in the literature concerning the ocrelizumab infusion. This connection was made based on the temporal relationship between the treatment and the onset of symptoms, as well as histological examination and ruling out other possible causes [13].

Although not many studies exist to support the link between this drug and psoriasiform dermatitis, there are reports in the literature about psoriasis vulgaris induced by rituximab (another monoclonal antibody directed against CD-20 expressing cells). The common target of these two drugs supports the association, the hypothesis being that the disruption of the T and B cell equilibrium and the increased susceptibility to viral or bacterial infections can lead to new-onset psoriasis [13]. Furthermore, there are reports of patients with rheumatoid arthritis or lupus who were treated with rituximab and later developed psoriasis [23].

In the existing papers, it is also described that the altered B cells that play the role of antigen-presenting cells can lead to excessive activation of T cells, thereby producing cutaneous immune-mediated diseases. Among these, palmoplantar pustulosis or oral lichen planus have been described [11]. However, given the macroscopical aspect of our patient’s lesions and his skin biopsy, these two possibilities were excluded.

In our case, the appearance of lesions in connection with the periodic infusion, the lack of other constant treatment, and the skin biopsy led to the conclusion that this is an ocrelizumab cutaneous adverse reaction.

Although unusual and not described in the existing literature in connection to ocrelizumab treatment, our patient’s skin biopsy revealed nummular disseminated eczema. As mentioned above, this is a chronic inflammatory cutaneous lesion whose pathophysiology is related to the dysfunction of the epidermal lipid barrier, xerosis, the colonization with Staphylococcus aureus and a modified immune response. Taking into consideration the effects of this monoclonal antibody on the immune response and the lymphocyte’s equilibrium, we can propose two possible causes for this association. First of all, the altered immune response can lead to an increased susceptibility to certain infections such as S. aureus or Candida albicans and, secondly, the immune disruption can alter the skin barrier, causing hypersensitivity and an irritant response to environmental factors.

Apart from the infusion-related reactions (IRR), which are due to the cytokine and other chemical mediators released, and from type I hypersensitivity reactions and infections, there are no other adverse cutaneous reactions described in the drug leaflet. The skin-related lesions that occur with the IRRs are pruritus, rash, urticaria, and erythema, but the period is usually 24 h after the infusion [12]. In contrast, our patients developed symptoms approximately a week after the administration of the drug, without any other identified causative factor.

With this ocrelizumab case, we signal a possible adverse reaction not described before in existing studies or in the drug leaflet. Through our observations we raise the clinician’s awareness of a new course of evolution that may influence patient compliance. Therefore, we must emphasize the possibility of these reactions, to inform the patient and prevent a negative outcome.

Following the results of these two presented cases, we can strongly affirm that it is of crucial importance to personalize the treatment to maximize the patient’s compliance. The physician must evaluate the individual’s characteristics, medical history and comorbidities and establish a plan of treatment based on all these data. There are many criteria described so far in the literature regarding algorithms for the choice of the best disease-modifying treatment. Nevertheless, the clinician must approach each case carefully and determine the drug of choice for each individual, based on clinical evolution, patient preference and adverse reactions that can occur during the treatment.

We believe that these reactions must be highlighted and taken into account when the decision to change the treatment or to switch it with another drug is being taken. Therefore, it is a matter of individual clinical judgement and patient compliance whether the treatment should be interrupted or not if these adverse cutaneous reactions appear. This is one of the questions raised by our study, having the purpose to challenge the clinicians to imagine new clinical scenarios and to develop new strategies for the maximization of the treatment efficacy.

## 5. Conclusions

We report two cases of drug-induced cutaneous lesions, one following Natalizumab treatment and one following Ocrelizumab infusions. Following the result of the skin biopsy, the temporal connection between the therapy and their issue and the lack of any other causative factor supports the causal relationship between them. Further studies are needed to explain the possible cutaneous adverse reactions and the mechanisms involved. This is important so that the clinician can predict the course of the treatment, discover ways to prevent adverse reactions, and firmly advise the patients for or against the continuation of the drug. We also want to emphasize the fact that the neurologist must always bear in mind the possibility of an adverse reaction, even after many years of well-tolerated treatment.

## Figures and Tables

**Figure 1 jcm-11-03702-f001:**
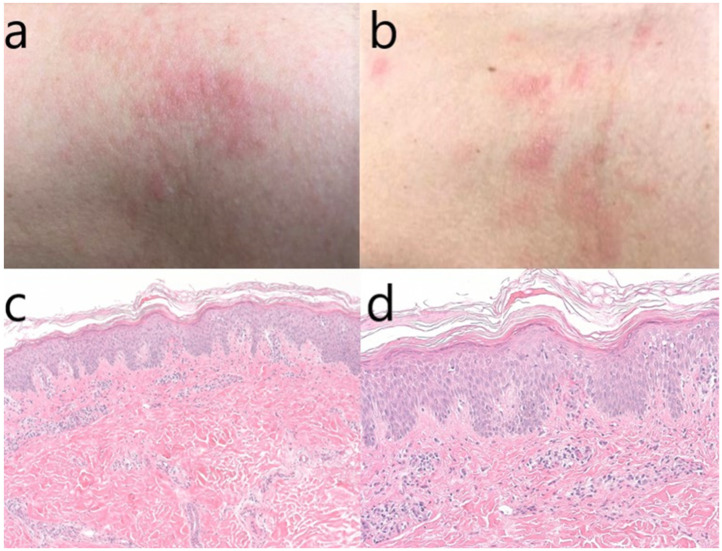
(**a**–**d**) Macroscopic aspect of the skin lesions from the neck and thorax (**a**,**b**); skin biopsy–hematoxylin & eosin stain, original magnification 100× (**c**); magnification 200× (**d**). We can observe hyperkeratosis, marked acanthosis (psoriasiform hyperplasia) with focal lymphocytic exocytosis, minimal epidermal edema, perivascular and interstitial lymphocytic infiltrate, which are histopathological signs compatible with subacute chronic spongiotic dermatitis.

**Figure 2 jcm-11-03702-f002:**
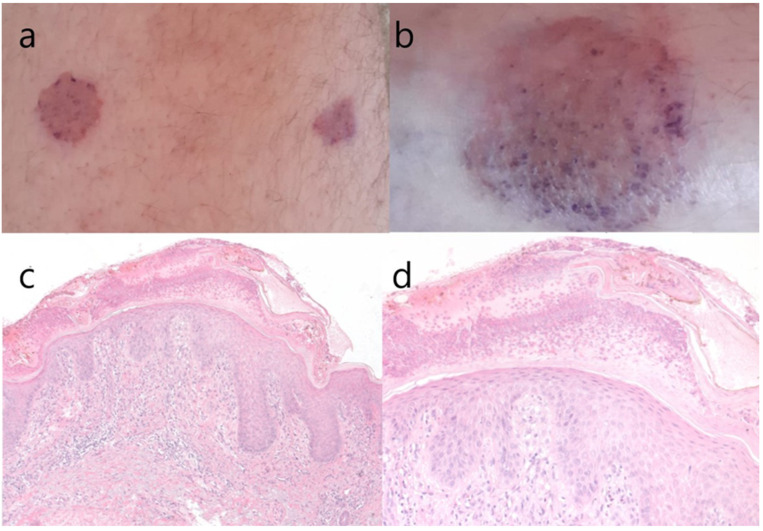
(**a**–**d**) Macroscopic aspect of skin lesion from the anterior thigh and the distal leg (**a**,**b**); hematoxylin & eosin stain, original magnification 100× (**c**); original magnification 200× (**d**). We can observe superficial vesicles with neutrophils, hyperkeratosis, epidermal hyperplasia with spongiosis, hypergranulosis (a pronounced granular cell layer) and exocytosis of lymphocytes, which are histopathological signs suggestive of nummular eczema or discoid eczema (clinically “silver dollar sized patches”).

## Data Availability

Not applicable.

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
