# Peer review of "Cutaneous Adverse Reactions Associated with Monoclonal Antibodies Treatment in Multiple Sclerosis: Case Reports and Short Literature Review"

_jcm, 2022, doi:10.3390/jcm11133702_

Round 1
Reviewer 1 Report
I believe that the data obtained on the two individual patients should be further investigated as it does not add anything new to what is already on the net.
Author Response
Please, see the attachment!

Reviewer 2 Report
Dear Authors,
the work is interesting, points one of the main side effects from systemic biological treatment. The molecular base of this problem is worth further clinical studies.
Author Response
Dear reviewer, we would like to thank you for your time and patience to review our paper! We finished the English-edited manuscript, and we hope is better now!
Reviewer 3 Report
1/ what do you mean by the term "urticaria erythema" ? It is not medical term ... is it urticaria ? or eythema ? Topical corticosteroids are not use to treat urticaria.
2/ why did you use oral antibiotics to treat disseminated ?
3/ did you perform allergological tests to excluded other causes of eczema ?
Reviewer 4 Report
The article describes 2 clinical cases about skin reactions to biological treatments for multiple sclerosis. The first case describes an urticarial reaction to natalizumab, that specific raction can be considered unusual (occurs after several months to start the treatment), even if present in the leaflet. The second case describes a late nummular skin reaction to ocrelizumab, that does not appear among the previously reported adverse effects for this drug.
In general, the article adds information that may be useful to practitioners dealing with multiple sclerosis and using monoclonal drugs.
Introduction.
The authors begin the introduction by stating that multiple sclerosis can have a progressive or relapsed course, however the bibliographic reference does not match with what has been said (line 43).
Case 1.
In the course of the account of the first clinical case the authors describe recurrent
episodes of urticaria, stating that these episodes resolve after a few days.
However, it is unclear whether the individual lesions (wheals) are fleeting,
this feature would suggest a histaminergic nature of the rash.
It is also not described whether the natalizumab infusion includes the use of
other ancillary drugs (such as corticosteroids, electrolytes, or acetaminophen),
or whether the patient uses drugs in the days following the infusion.
It would be interesting to know if the room of the administration is entirely latex free
and if the patient has been tested for specific IgE against seasonal allergens and latex.
Round 2
Reviewer 1 Report
The work seems to me not yet of scientific impact, but it is really detailed how you have processed the comments and adapted the text accordingly
Reviewer 3 Report
1. ref 24 - in my opinion the websites shud not be a reference